# Optimizing the Microstructure and Corrosion Resistance of BDD Coating to Improve the Service Life of Ti/BDD Coated Electrode

**DOI:** 10.3390/ma12193188

**Published:** 2019-09-28

**Authors:** Xin ru Lu, Ming hui Ding, Lu Zhang, Zhi liang Yang, Yao Lu, Wei zhong Tang

**Affiliations:** Institute for Advanced Materials and Technology, University of Science and Technology Beijing, Beijing 100083, China

**Keywords:** Ti/BDD coated electrode, microstructure, corrosion resistance, the service life

## Abstract

The short service life of the Ti/BDD coated electrode is the main reason that limits its practical use. In this paper, the effect of structural change on the service life was studied using Ti/BDD coated electrodes prepared with the arc plasma chemical vapor deposition (CVD) method. It was found that the microstructural defects and corrosion resistance of BDD coatings were the main factors affecting the electrode service life. By optimizing the process parameters in different deposition stages, reducing the structural defects and improving the corrosion resistance of the BDD coating were conducted successfully, which increased the service life of the Ti/BDD coated electrodes significantly. The lifetime of the Ti/BDD samples increased from 360 h to 655 h under the electrolysis condition with a current density of 0.5 A/cm^2^, with an increase of 82%.

## 1. Introduction

As an excellent anode electrode material, the boron-doped diamond (BDD) electrode has been extensively studied and reported in the literature on its application in the wastewater treatment industry [1,2]. BDD coatings are generally deposited on substrates of silicon (Si), tantalum (Ta), titanium (Ti), and niobium (Nb) is used as a coating electrode [3,4]. The Ti/BDD coated electrode combines the advantages of Ti and BDD materials, which has always been a research hotspot [5,6]. Ti/BDD coated electrodes have broad application prospects in electrochemistry including: (1) disinfection and treatment of water; (2) electrochemical sensors; and (3) electrosynthesis of organic and inorganic compounds, etc. [7,8,9]. However, Ti/BDD coated electrodes have an ordinarily short service life, which limits its practical industrial application [10,11,12].

Previous studies have found that coating delamination was the primary failure mode of Ti/BDD coated electrodes [10,11,12,13]. It is generally believed that corrosion of the TiC layer at the interface of the substrate/coating, quality and adhesion of the BDD coating, high residual stress produced in the chemical vapor deposition process, and the corrosion of the diamond phase under high potential are the possible causes of electrode failure [11,13,14,15]. To improve the service life of Ti/BDD coated electrodes, some attempts have been carried out by different researchers in terms of modifying the interface structure of the coating/substrate. Xue et al. [11,15] improved the service life of a Ti/BDD electrode significantly by adding CH_2_(OCH_3_)_2_ to the reaction atmosphere of CH_4_ and H_2_, and believed that the increase of diamond nucleate density and growth rate reduced the thickness of the TiC layer. Yuan et al. [16] prepared Ti/Si/BDD electrodes by depositing a Si layer on a Ti substrate primarily, which further improved the electrode life. It was believed that the main reason was that the Si intermediate layer restricted the formation of the TiC layer. Guo et al. [13] prepared a Ti/BDD electrode sample with a multilayer structure of a Ti/TiC/diamond + amorphous carbon/diamond using a two-stage substrate temperature and suggested that the multilayer structure increased the adhesion of the diamond film onto the substrate significantly, which improved the electrode life. Sun et al. [17] improved the stability of Ti/BDD electrodes by increasing the B-doping content in the initial growth stage. It was believed that the increase of diamond nucleation rate reduced the TiC growth rate and promoted a more compact interface structure of the coating/substrate. Furthermore, Wei et al. [14] found that the B-doped content had a significant impact on the quality and adhesion of the BDD film, and the appropriate amount of B-doped could improve the stability of the Ti/BDD electrode.

Based on our recent research results, we found that the failure process of the Ti/BDD electrodes could be divided into two stages. The reason for the coating delamination in the first stage was mainly due to the existence of pore type defects in the BDD coatings, while the reason for the coating delamination in the second stage was mainly related to the corrosion holes formed in the electrolytic process. Therefore, possible ways to improve the service life of Ti/BDD electrodes were summarized [18,19].

In this paper, to improve the stability of the Ti/BDD electrodes, the process parameters at different deposition stages of BDD coatings were modified to optimize the interface microstructure of the coating/substrate, reduce the structural defects in the BDD coating, and improve the corrosion resistance of the BDD coating. This will provide ideas and data support for promoting the practical application of Ti/BDD coated electrodes.

## 2. Materials and Methods

### 2.1. Preparation of Samples

Ti plates were cut into 18 × 15 × 1 mm samples and mechanically ground with sandpapers from #200 to #800 successively. BDD coatings were prepared by using a high current extended direct current arc plasma CVD system [20]. Before deposition, surface pretreatment by mechanical grinding with 0.5 μm diamond powder was adopted to promote nucleation. The reactive gas used was a mixture of argon (Ar), hydrogen (H_2_), and methane (CH_4_). Trimethyl borate (B(OCH_3_)_3_) was used as the source for elemental boron. Deposition of the BDD coatings was carried out in two stages. First, during the nucleation stage, a higher methane concentration was used to increase the diamond nucleation density on the substrates, and then during the growth stage, the methane concentration was lowered to improve the quality of the BDD coatings. The deposition time was 2 h for the nucleation stage and 8 h for the growth stage, respectively. Temperature, pressure, the volume flow rate of Ar and H_2_ remained constant during deposition. Specific parameters were 750 °C, 500 Pa, 1800 sccm (standard-state cubic centimeter per minute), and 100 sccm, respectively.

Considering that the interface microstructure of the coating/substrate is mainly affected by the nucleation process, and the structure and corrosion resistance of the BDD coating is closely related to the doping amount of B impurities, the optimization process was carried out in three stages.

(1) Study the influence of CH_4_ flow rate on the interface microstructure of the substrate/coating in the nucleation stage. Detailed deposition parameters are given in Table 1.

(2) Study the effects of B impurity flow rate on the structure and corrosion resistance of BDD coatings during deposition. Detailed deposition parameters are given in Table 2.

(3) Based on the results of stages (1) and (2), obtaining the optimized preparation parameters was used as process-1. On this basis, we adopted the gradient B-doping method to further optimize the structure and corrosion resistance of BDD coating, which was process-2. During process-2, the B source flow rate was 0.4 sccm in the early stage of deposition for 8 h and then changed to 0.1 sccm in the late stage of deposition for 2 h. Detailed deposition parameters of two processes are given in Table 3.

### 2.2. Characterization of Ti/BDD Samples

Surface and cross-section morphology of the samples was examined using an FEI Quanta 450 field emission scanning electron microscope (FE-SEM) (FEI Company, Hillsboro, OR, USA). Structure of the BDD coatings was characterized by Raman spectroscopy conducted using a Raman 2000 spectrometer (HORIBA Ltd., Paris, France).

The electrochemical performance of the study samples characterized by cyclic voltammetry (CV) curves was performed on a CS2350 electrochemical workstation with a classical three-electrode configuration (Wuhan corrTest Instruments Co., Ltd., Wuhan, China). Ti/BDD samples with an exposed area of 1 cm^2^ served as the working electrode. A platinum plate was used as the counter electrode. A saturated calomel electrode (SCE) (INESA Scientific Instrument Co., Ltd., Shanghai, China) was used as the reference electrode. The measurements were conducted in 0.5 mol/L H_2_SO_4_ electrolyte at a scanning rate of 0.1 V/s and the potential ranged from −1.5 to 3 V (vs. SCE).

Accelerated life tests (ALT) on the Ti/BDD samples were carried out in 1 mol/L H_2_SO_4_ electrolyte by employing a two-electrode system. A TRADEX MPS 302 power source (Beijing TRADEX Electronic Technology Co., Ltd., Beijing, China) was used to provide constant current densities of 0.5 and 1 A/cm^2^ to the electrolytic cell. Ti/BDD coated electrodes and a stainless steel plate were used as the anode and cathode, respectively. When the cell voltage rose abruptly, the BDD electrode was considered to have failed.

## 3. Experimental Results and Discussions

### 3.1. Effect of the CH_4_ Flow Rate during the Nucleation Stage on the Service Life of Ti/BDD Samples

Figure 1 shows the morphology of the BDD coatings on the Ti substrate at different CH_4_ flow rates after a deposition of 2 h. When the CH_4_ flow rates were 5 sccm and 6 sccm, we could see that abundant fine diamond nuclei covered the surface of the samples, but there remained a considerable amount of interstitial space between the diamond grains, showing that no continuous diamond coating formed. When the CH_4_ flow rate was 7 sccm, the nucleation density further increased, but there were still a few voids on the surface. When the CH_4_ flow rate was higher than 8 sccm, continuous fine grain coatings formed on the surface, but large grains existed locally. The above phenomenon is mainly because the diamond nucleation can only take place after the formation of a carbide layer on the Ti substrate surface. At the initial stage of deposition, there was competition between the carbide formation and the diamond nucleation [21,22]. Uniform nucleation of the diamond phase was only possible when the C concentration reached the critical concentration of nucleation. When the concentration of CH_4_ was too low, the necessary super-saturation concentration of the C was difficult to reach on the Ti substrate surface at the early stage of deposition, which delayed the nucleation of the diamond phase and reduced the nucleation density. Only when the concentration of CH_4_ was high enough, was the re-dissolution of the diamond phase nuclei hindered, and the nucleation process could proceed smoothly. However, when the CH_4_ flow rate was too high, some nucleated preferentially diamond grains overgrew, resulting in uneven diamond particles contained in the coatings, as shown in Figure 1d–f.

Figure 2 shows the cross-section morphology of the Ti/BDD samples obtained by changing the CH_4_ flow rate during the nucleation stage and then deposition for 8 h with a CH_4_ flow rate at 3 sccm. As can be seen from Figure 2, when the CH_4_ flow rate was 5 sccm or 6 sccm during the nucleation stage, the grain size at the interface of the substrate/coating was relatively large. Due to the low concentration of CH_4_, a rough and porous interface microstructure formed, as shown in Figure 2a,b. With the increase in the CH_4_ flow rate, the interfacial microstructure became compact. When the CH_4_ flow rate was higher than 8 sccm, the BDD coating microstructure changed obviously. It can be observed that there was a fine grain layer on the surface of the substrate, and the top of the BDD coating was a columnar structure.

Figure 3 depicts a comparison of the accelerated life tests (ALT) life of the Ti/BDD electrodes prepared at different CH_4_ flow rates during the nucleation stage under current densities of 1 A/cm^2^ and 0.5 A/cm^2^. From the test results, increasing the CH_4_ flow rate in the nucleation period, the ALT life of the BDD electrode first increased and then decreased. When the CH_4_ flow rate was near 8 sccm, the ALT life of the electrode was the highest.

### 3.2. Effect of the B Source Flow Rate on the Service Life of Ti/BDD Samples

Previous studies have shown that the B-doped amount has a significant impact on the morphology, structure, and quality of the BDD films [14,23,24] Therefore, this section will study the effect of B source flow rates on the service life of Ti/BDD coated electrode.

Figure 4 shows the micro-morphology of the electrode samples prepared under different B-source flow rates after 10 h of electrolysis. It should be mentioned that they are similar to the morphology of BDD coatings before electrolysis. According to Figure 4, by increasing the B-doped content, the BDD grain size decreased. When the flow rate of B source was 0.1 sccm, the grain size was significantly larger than 1 μm. When that was between 0.2–0.4 sccm, the grain size was about 1 μm. The variation of the grain size indicates that increasing the B-doped content can refine the grain size of the BDD coatings, which is consistent with the reported results [25].

From Figure 4a–c, when the B source flow rate was less than 0.4 sccm, no obvious corrosion occurred on the grain surface after electrolysis for 10 h. When the B source flow rate was 0.6 sccm, slight corrosion could be found at the grain boundaries after electrolysis, as shown in Figure 4d. When the B flow rate was 0.8 sccm, abundant corrosion holes appeared at grain boundaries and grain planes after electrolysis. From the above results, we can conclude that the corrosion resistance of BDD coating decreased with the increase in the B-doped content. It can be explained that B entering into the diamond structure caused lattice distortion, increasing the defects in the diamond grains. In particular, B atoms tended to segregate at the grain boundaries, increasing the non-diamond phase content at the grain boundaries [24,26]. It is generally believed that these defective areas vulnerably corroded. In the electrolysis process, electrolytes can easily invade along these preferential corrosion sites, thereby reducing the electrode life.

Figure 5 shows the results of the ALT life of the Ti/BDD coated electrodes with different B-doped content at constant current densities of 1 A/cm^2^ and 0.5 A/cm^2^. It can be seen from the results in Figure 5 that under the two current densities, the trend of the ALT life of electrode samples varied with the B-doping content similarly. The ALT life of the electrodes first increased and then decreased with the increase of the B source flow rate. The Ti/BDD electrode sample had the highest ALT life when the B source flow rate was 0.4 sccm. We noticed when the B source flow rate was less than 0.4 sccm, that by increasing the B source flow rate, the electrode life increased. This is mainly because increasing the B-doping content increased the nucleation rate and density of the diamond, and the interface structure of coating/substrate became compact. Such a result is consistent with the report of improving the electrode life by increasing the B-doping content during the nucleation period [17]. However, when the B flow rate was greater than 0.4 sccm, with the increase of the B flow rate, the electrode life decreased. As shown in Figure 4, this may be due to the decreased corrosion resistance of the BDD coating with the increase in the B-doping content, thus reducing the electrode life.

### 3.3. Adopting the Gradient B-Doping Method to Improve the Service Life of Ti/BDD Coated Electrode

From the above results, it can be concluded that the CH_4_ flow rate during the nucleation period has an important influence on the interface structure of the Ti/BDD coated electrodes, while the B-doped content has a significant influence on the structure and corrosion resistance of the BDD coatings. Under appropriate conditions, the Ti/BDD coated electrode has a multi-layer structure with substrate + fine-grain layer + columnar crystal layer, and the BDD coating has good corrosion resistance. In this section, the electrode samples were first prepared by the optimized process based on the results in Section 3.1 and Section 3.2, named as the process-1 sample, and then the electrode samples were prepared by adopting the gradient B-doping method to further optimize the structure and corrosion resistance of the BDD coating, named the process-2 sample. Specific process parameters are shown in Table 3.

#### 3.3.1. Characterization of Electrode Samples Prepared by Two Processes

Figure 6 shows the surface and cross-section morphologies of the Ti/BDD electrode samples prepared by both process-1 and process-2. As shown in Figure 6a, after 8 h of deposition, the grain size of the BDD coating was about 1 μm. The morphology of the crystal was dominated by the triangular (111) plane, and the grain shape was mainly octahedral, but there were inter-crystalline gaps on the surface. After further deposition for 2 h without changing the process parameters, the process-1 sample was obtained, as shown in Figure 6b. Comparing Figure 6a,b, with the increase of deposition time, the grain size and morphology changed slightly. While adopting the gradient B-doping method, the process-2 sample was obtained, as shown in Figure 6c. Compared with the process-1 sample, the grain size of the BDD coating increased slightly. This is mainly due to the fact that the growth rate of BDD increases with the decrease of B content under the condition of high B doping. Simultaneously, the grain shape was mainly polyhedral twin grains. The existing literature showed that the variation of B-doping content could significantly change the morphology and grain orientation of the BDD coating [24,25,27]. By comparing Figure 6d,e, on the top layer of the BDD coating of the process-2 sample, the grain orientation changed and the pyramidal hillocks diminished. In the columnar structure layer, the grain size increased, the number of small grains decreased, and the grain boundaries were reduced.

Figure 7 shows the Raman spectra of the BDD coatings prepared by two processes. We could see that the Raman spectra of the two samples showed distinct characteristics. The spectra of the process-1 sample exhibited typical characteristics of the heavily doped BDD coating [28]. Two broad peaks at around 500 cm^−1^ and 1200 cm^−1^ were suggested to be due to amorphous diamond. The asymmetry and seriously deterioration of the characteristic diamond peak at 1332 cm^−1^ were attributed to the Fano-effect caused by heavily B-doping. For the spectra of the process-2 sample, the peak of the diamond centered at 1332 cm^−1^ became sharper and stronger, and the broadband centered around 1500 cm^−1^ appeared. Generally, the peak around 1500 cm^−1^, assigned to disordered graphite, diminishes with an increase in the B/C ratio in the reaction gas [29]. On the other hand, the peak position at around 1332 cm^−1^ and 500 cm^−1^ shifted to a high wavenumber. The downshift of these two peaks with an increase in the B/C ratio have been extensively reported [23,28,30]. All of these indicated that the B-doping content decreased in the BDD coating of the process-2 sample. According to the literature [31], the Raman peak position near 500 cm^−1^ can be used to estimate boron concentration in the BDD films. The B concentration of the process-1 and process-2 samples were estimated as an order of 2.3 × 10^21^/cm^3^ and 5.2 × 10^20^/cm^3^, respectively.

Figure 8 shows the CV curves of the Ti/BDD electrode samples prepared by the two processes in 0.5 mol/L H_2_SO_4_. From Figure 8, both samples have typical electrochemical properties of BDD electrodes: a wide potential window, high oxygen evolution potential, and low background current. A careful comparison shows that the potential window of the process-2 sample increased slightly, which can be attributed to the decrease of B-doping content on the surface of the BDD coating.

#### 3.3.2. The ALT Results of Ti/BDD Coated Electrodes Prepared by Two Processes

Figure 9 shows the ALT life of Ti/BDD coated electrodes prepared under two processes. The results showed that the ALT life of the electrode significantly improved after adopting the gradient B-doping method. At the current density of 1 A/cm^2^, the ALT life of the Ti/BDD electrode increased from 147 h to 196 h, with an increase of 33%. At 0.5 A/cm^2^, the ALT life of the Ti/BDD electrode increased from 360 h to 655 h, with an increase of 82%.

Figure 10 shows the morphology of electrode samples prepared by the two processes after different electrolysis time during the ALT process. As can be seen from Figure 10, the corrosion of the BDD coating occurred on both of the Ti/BDD coated electrodes. For the process-1 sample, after 120 h of electrolysis, only a small number of holes appeared at the grain boundary on the surface of the BDD coating. After 300 h of electrolysis, the corrosion holes at the grain boundaries increased, and the corrosion seemed to be more severe at the small grains. Regarding the process-2 sample, after 120 h of electrolysis, only slight corrosion occurred at the grain boundaries on the BDD coatings. After 300 h of electrolysis, corrosion further intensified, but no visible corrosion holes appeared. After 600 h of electrolysis, the grains became irregular, and a few corrosion holes appeared locally. During the ALT process of the process-2 sample, the small number of corrosion holes that emerged only in the later stage of electrolysis can be explained by the reduction of structural defects and the improvement in the corrosion resistance of the BDD coating, according to the results presented in Section 3.2 and Section 3.3.1. Therefore, the service life of the process-2 samples was significantly improved.

### 3.4. Mechanisms Analysis of Improving the Service Life of Ti/BDD Coated Electrode

During the preparation process of the Ti/BDD samples, a carbide layer was preferentially formed on the surface of the Ti substrate, then the nucleation and growth of diamond occurred. It was difficult to completely avoid the structural defects in the BDD coating, as shown in Figure 2. In electrolysis, the corrosion holes formed on the BDD coating, as shown in Figure 10. During the ALT process, these defects would lead to the electrolyte intruding into the BDD coating and corroding the bonding sites between the substrate and coating. The titanium carbide layer is unstable and decomposes easily at high potential [15,32,33], which results in the BDD coating, gradually peeling off. The reason for the coating delamination in the first stage was mainly the existence of pore type defects in the BDD coatings, while the reason for the coating delamination in the second stage was mainly related to the corrosion holes formed during the electrolytic process. Therefore, the preparation and failure process of the Ti/BDD coated electrode can be seen in Figure 11 (a→b→c→d→e→f).

From Section 3.1, when the CH_4_ flow rate was too low in the nucleation stage, the diamond nucleation density was insufficient, leading to the formation of a porous interface structure of the coating/substrate. Not difficult to imagine that, when the electrolyte penetrated the BDD coating along with structural defects such as grain boundary pore or corrosion holes, this porous interface structure was not effective in preventing the electrolyte from eroding the substrate, resulting in the delamination of the BDD coatings. In contrast, by increasing the CH_4_ concentration, the interface structure of the coating/substrate became compact due to the formation of a dense fine grain layer, which could delay the substrate corrosion and improve the electrode life. However, the increase of CH_4_ content would lead to the increased content of the non-diamond phase in the fine-grain layer [34]. The non-diamond phase with a low corrosion resistance would dissolve preferentially and accelerate the coating delamination during the ALT process [35]. Therefore, when the CH_4_ flow rate is too high, the electrode life can also be reduced.

By optimizing the nucleation process, the interface microstructure of the coating/substrate became compact, which could delay the decomposition of the carbide layer during the ALT process and thus improve the electrode life. The schematic diagram of the structural change is shown in Figure 11b→b^1^. In previous studies [11,16,17], the methods of introducing other transition layers (such as si transition) or increasing the nucleation density to improve the service life of the Ti/BDD electrodes are based on the design idea of modifying the interface structure of the coating/substrate.

During the CVD process of the preparation diamond, the crystal morphology and orientation are closely related to the growth parameter *α* [36,37]. The *α* was defined as *α* = √3 *ν*_100_/*ν*_111_, which describes the ratio of the growth rates on (100) and (111) faces and depends primarily on the gas composition and substrate temperature. For polycrystalline diamond films, the growth parameter *α* has an essential influence on the film texture and the crystal morphology, and finally determines the morphology, structure, and defects of the diamond films [38,39]. Without changing the process parameters, the growth parameters are approximately constant. On one hand, with grains growing upward, the octahedral grains may form a “canyon” intergranular environment, which makes it difficult for reaction groups to reach, thus resulting in void defects. On the other hand, the vertical growth of columnar crystal layers would result in the grain boundary regions being approximately perpendicular to the sample surface. The structure of the BDD coating under this growth condition can be shown in the schematic diagram of Figure 11c,c^1^. Structural defects can cause electrolytes to invade the coating prematurely, which is the main factor causing the early delamination of the coating, as shown in Figure 11d. The grain boundaries corroded preferentially and formed corrosion holes in the electrolysis process, as shown in Figure 10a,b. Such an approximately perpendicular grain boundary could cause the electrolyte to reach the coating/substrate interface quickly, which is the main cause of coating delamination in the later process, as shown in a corrosion failure mode in Figure 11e.

Figure 11c^2^ shows the BDD coating structure of the process 2 samples. The gradient boron doping method was adopted to reduce the B flow rate at the later growth stage. Compared with the process 1 samples, the surface and cross-section morphology had significant changes. This was mainly due to the value of *a* changed, leading to a varied grain orientation and an increase in the grain size. On one hand, the structural defects formed in early growth were buried deeper due to the linear direction of the grain boundaries changed. On the other hand, some tiny grains disappeared due to the increase in the grain size, thus reducing the structural defects at the grain boundaries. In addition, the corrosion resistance of the BDD coating was improved due to the decrease of B doping at the later growth stage. All changes above would help delay the electrolyte intrusion into the BDD coatings. These conclusions were verified by the morphological changes of the failure process shown in Figure 10, so that the service life of the electrode was significantly improved.

## 4. Conclusions

In this paper, the structural changes of Ti/BDD coated electrodes during the preparation process were studied. It was found that the structural defects in the BDD coatings were the main factors affecting the lifetime of the electrodes. By optimizing the process parameters at different stages of BDD deposition, the following conclusions were drawn:The nucleation process has an important effect on the coating/substrate interface microstructure of the Ti/BDD coated electrode. Optimizing the nucleation process and reducing the hole defects at the coating/substrate interface can significantly improve the service life of a Ti/BDD coated electrode.B-doping content has a significant effect on the structure and corrosion resistance of BDD coating. The low B-doping content made the structure of the BDD coating not compact and decreased the life of the electrode. The high B-doping content caused the grain boundary defects of the BDD coatings to increase and the corrosion resistance to decrease, which greatly reduced the service life of the Ti/BDD coated electrodes.By adopting the gradient B-doping method, the value of the growth parameter a of the BDD coating changed at the later deposition stage, which led to a change in the grain orientation, the structural defects in the coating to diminish, and significantly improved the service life of the electrode samples.

## Figures and Tables

**Figure 1 materials-12-03188-f001:**
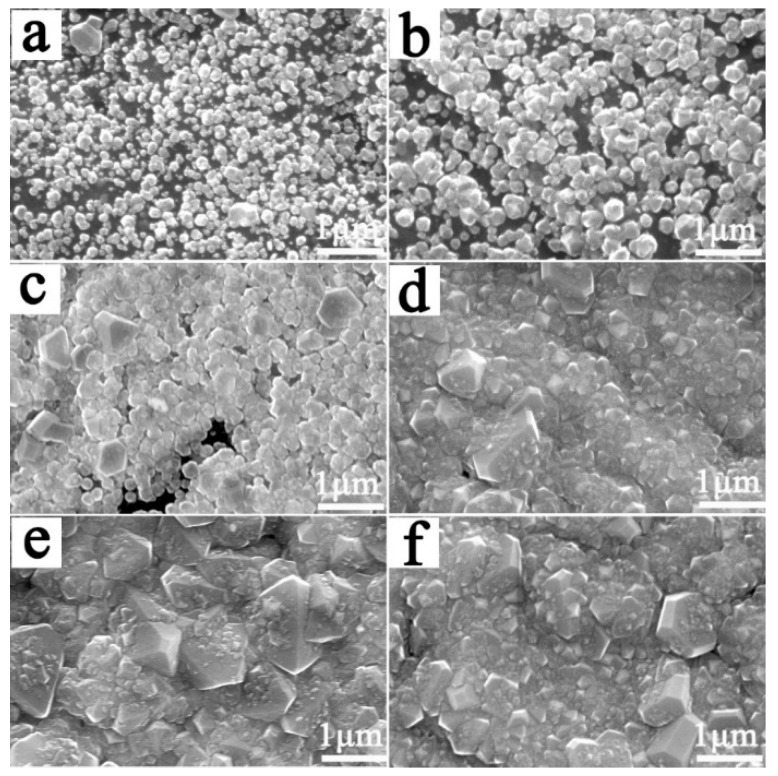
Morphology of Boron doped diamond (BDD) coatings deposited for 2 h at different CH_4_ flow rates:(**a**) 4 sccm; (**b**) 6 sccm; (**c**) 7 sccm; (**d**) 8 sccm; (**e**) 10 sccm; (**f**) 14 sccm.

**Figure 2 materials-12-03188-f002:**
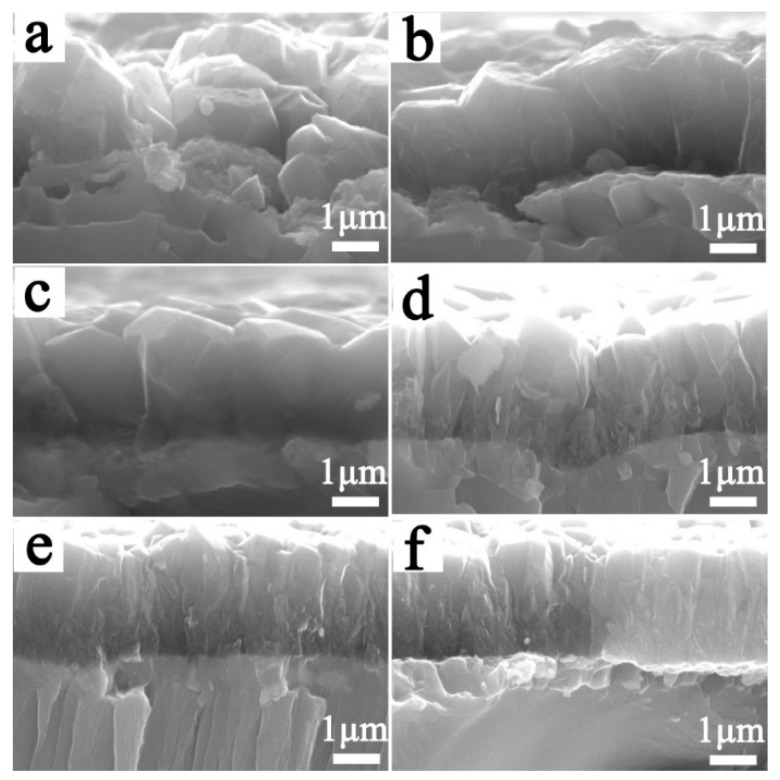
Cross-section morphology of the Ti/BDD samples prepared with a CH_4_ flow rate of (**a**) 5 sccm; (**b**) 6 sccm; (**c**) 7 sccm; (**d**) 8 sccm; (**e**) 10 sccm; and (**f**) 14 sccm in nucleation stage.

**Figure 3 materials-12-03188-f003:**
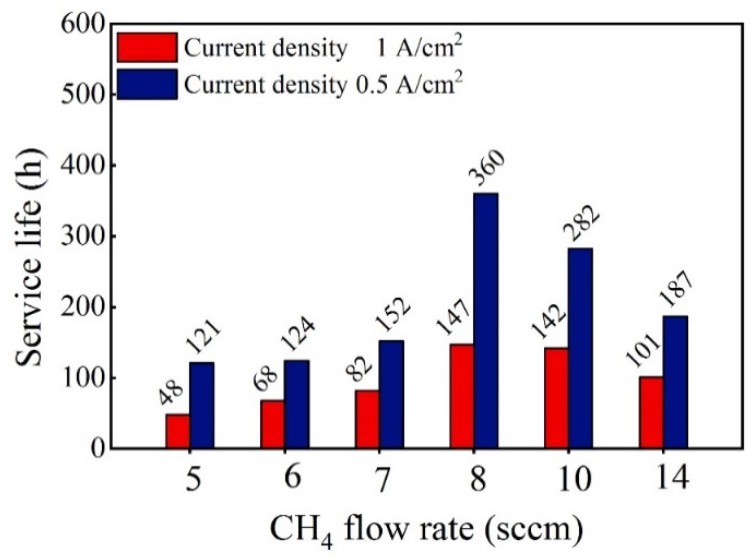
The effect of the CH_4_ flow rate during the nucleation stage on the accelerated life tests (ALT) life of Ti/BDD electrodes.

**Figure 4 materials-12-03188-f004:**
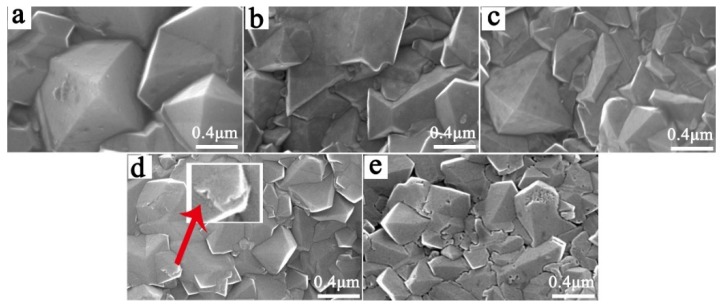
The surface morphology of the samples prepared at different B source flow rates: (**a**) 0.1 sccm; (**b**) 0.2 sccm; (**c**) 0.4 sccm; (**d**) 0.6 sccm; and (**e**) 0.8 sccm after 10 h of electrolysis. Electrolytic conditions: 1 A/cm^2^ current density. The illustration in (**d**) is a magnified image at the grain boundaries.

**Figure 5 materials-12-03188-f005:**
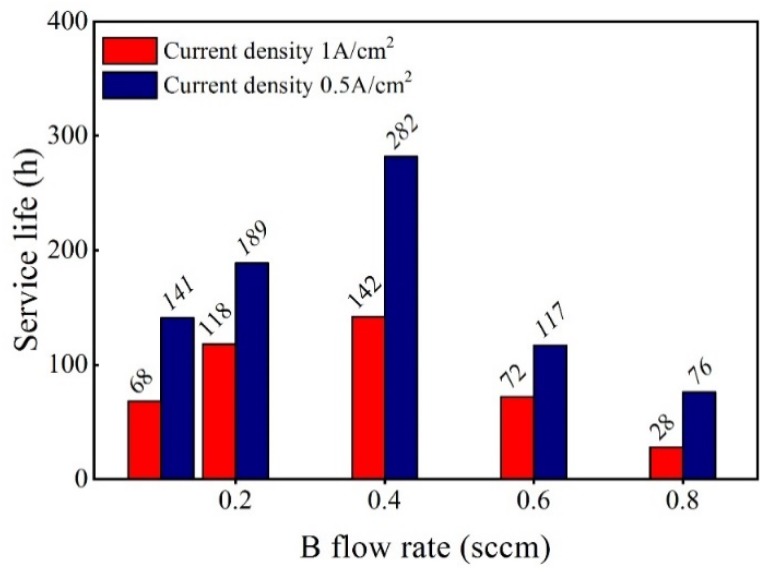
The effect of the B source flow rate on the ALT life of the Ti/BDD coated electrodes.

**Figure 6 materials-12-03188-f006:**
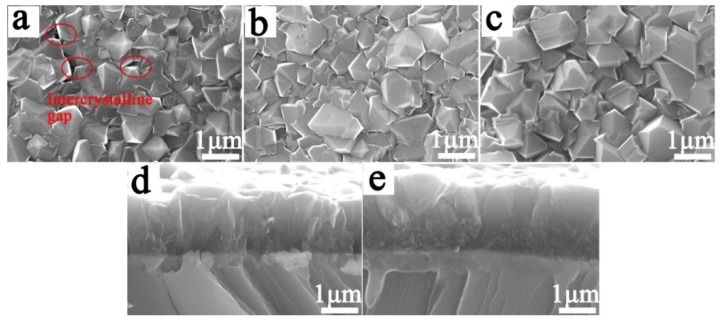
Morphologies of the Ti/BDD coated electrodes prepared under different process conditions. Surface morphology of (**a**) 8 h growth sample; (**b**) process-1 sample; and (**c**) process-2 sample. Cross-section morphology of the (**d**) process-1 sample and (**e**) process-2 sample.

**Figure 7 materials-12-03188-f007:**
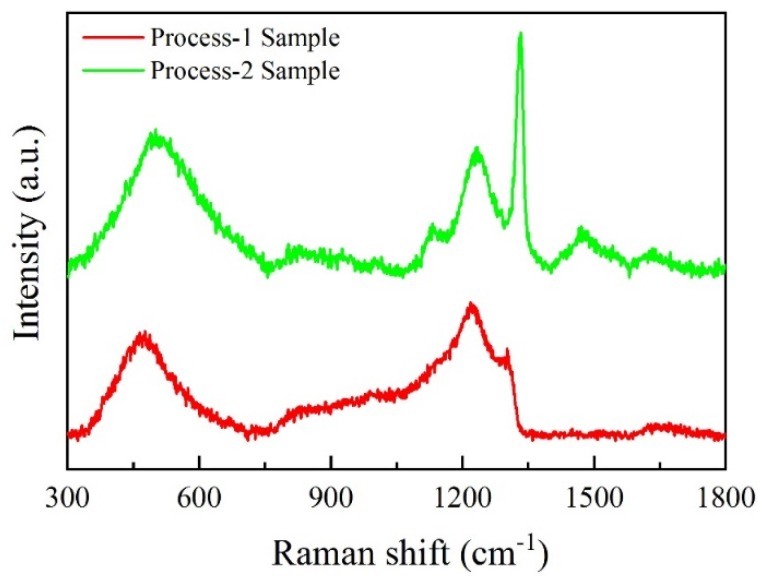
Raman spectra of the BDD coatings prepared by two processes.

**Figure 8 materials-12-03188-f008:**
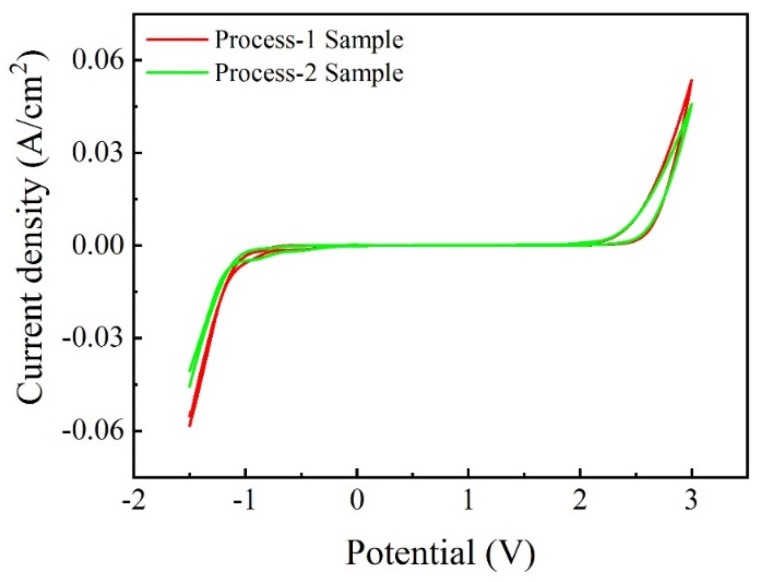
Cyclic voltammetry (CV) curves of Ti/BDD coated electrodes prepared by two processes.

**Figure 9 materials-12-03188-f009:**
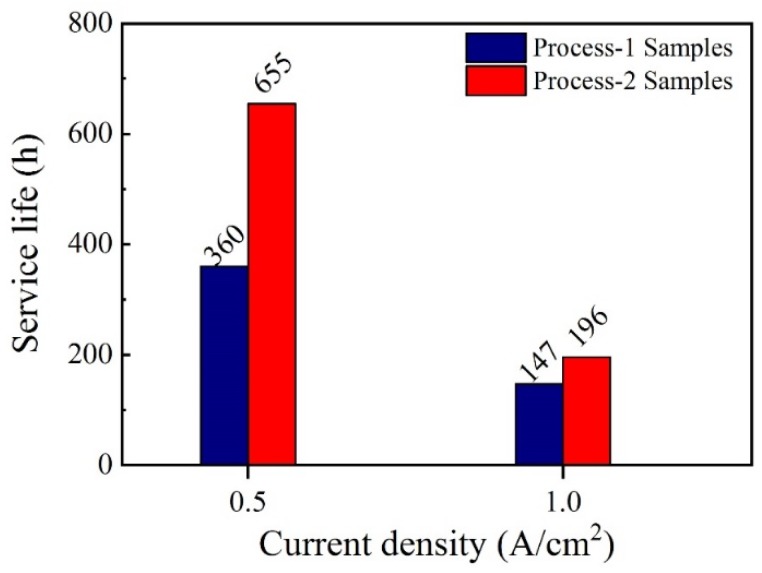
The ALT life of the Ti/BDD coated electrodes prepared by the two processes at different current densities in electrolysis.

**Figure 10 materials-12-03188-f010:**
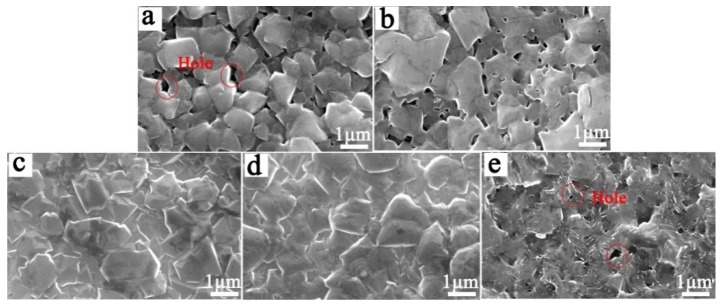
The morphology of the BDD coating of two samples after electrolysis of different times in 0.5 mol/L H_2_SO_4_ at a current density of 0.5 A/cm^2^. Process-1 sample: (**a**) 120 h; (**b**) 300 h. Process-2 sample: (**c**) 120 h; (**d**) 300 h; (**e**) 600 h.

**Figure 11 materials-12-03188-f011:**
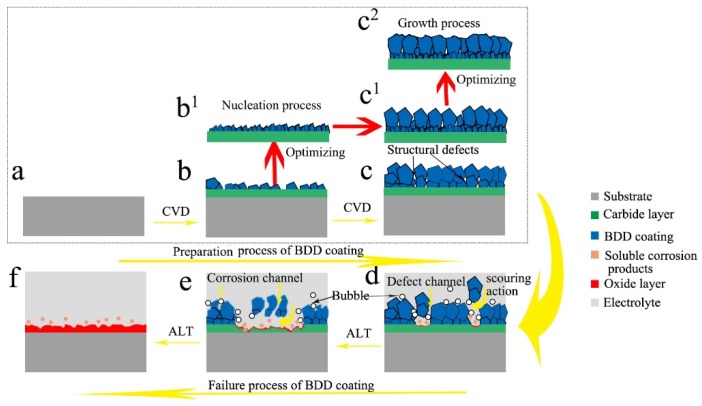
The preparation and failure process of the Ti/BDD coated electrode. The sample: (**a**) before deposition; (**b**) after the nucleation process; (**b^1^**) after optimizing the nucleation process; (**c**)as-depostied without any optimizing processes; (**c^1^**) as-depostied with optimizing the nucleation process; (**c^2^**) as-depostied with optimizing the nucleation process and adoting the gradient boron doping method; (**d**) in the first stage of the failure process; (**e**) in the second stage of the failure process; (**f**) completely failed.

**Table 1 materials-12-03188-t001:** Deposition parameters of the variation of CH_4_ flow rate in the nucleation stage.

Gas	The Volume Flow Rate (sccm)	
	The Nucleation Stage for 2 h	The Growth Stage for 8 h
CH_4_	5, 6, 7, 8, 10, 14	3
B Source	0.4	0.4

**Table 2 materials-12-03188-t002:** Deposition parameters of the variation of B source flow rate.

Gas	The Volume Flow Rate (sccm)	
	The Nucleation Stage for 2 h	The Growth Stage for 8 h
CH_4_	10	3
B Source	0.1, 0.2, 0.4, 0.6, 0.8	

**Table 3 materials-12-03188-t003:** Deposition parameters of the two processes.

Gas		The Volume Flow Rate (sccm)	
		The Nucleation Stage for 2 h	The Growth Stage for 8 h
Process-1	CH_4_	8	3
	B Source	0.4	0.4
Process-2	CH_4_	8	3
	B Source	0.4	0.4 (for 6 h) and 0.1(for last 2 h)

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
