# Peer review of "Optimizing the Microstructure and Corrosion Resistance of BDD Coating to Improve the Service Life of Ti/BDD Coated Electrode"

_materials, 2019, doi:10.3390/ma12193188_

Round 1

Reviewer 1 Report

The subject of the paper concerns the production of technological surface layers.
Very interesting paper.
The paper was written correctly.
The accepted methodology of research is correct.
The language of the paper is satisfactory.
The authors drew proper conclusions from their test results.
The paper meets the journal's requirements and can be published.

Author Response

Thank the reviewer for the comment

Reviewer 2 Report

The paper results are supported by the relevant discussion, however for paper better understanding for more readers it would be nice to describe in few words in the Introduction the application of the Ti/BDD.

Author Response

Thank the reviewer for the comments. In accordance with the reviewer’s suggestion, we added a description of the related applications of Ti/BDD electrodes and other references in the introduction section of the revised manuscript. The relevant part is as follows:

Ti/BDD coating electrodes have broad application prospects in electrochemistry, including: (1) disinfection and treatment of water, (2) electrochemical sensors, (3) electrosynthesis of organic and inorganic compounds, etc[7-9].

Reviewer 3 Report

The manuscript is a clear, reasonably self-contained presentation of the material, giving adequate reference.

A work of high scientific level with some minor changes to be made!

In order to be able to better observe the processes that occur, cyclic voltammetry must be performed at lower scanning rate. Thus, the difference between the two types of samples analyzed can be highlighted.

Author Response

Thank the reviewer for the comment. As the reviewer stated, the near steady-state polarization curve performed at lower scanning rate can show the surface state changes of the electrode in more detail. However, due to the wide potential range in the scanning process, the lower scanning rate often took a longer time, which may lead to more serious surface polarization. For the investigation of potential window and background current of BDD electrode materials, researchers mostly used fast scanning rates, such as 50 mV/s or 100 mV/s. In order to compare with the other reports, the scanning rate of 100 mV/s was adopted in this paper.